# Chagas prevention and control in an endemic area from the Argentinian Gran Chaco Region: Data from 14 years of uninterrupted intervention

Diego Weinberg[1], Maria Florencia Casale[1], Rosa Graciela Cejas[2], Rafael Hoyos[2], María Victoria Periago [1,3]*, Elsa Segura[3†], Marcelo Claudio Abril[1]

1 Fundación Mundo Sano, Buenos Aires, Argentina, 2 Fundación Mundo Sano, Añatuya, Santiago del Estero, Argentina, 3 Consejo Nacional de investigaciones Científica y Técnicas (CONICET), Buenos Aires, Argentina

☯ These authors contributed equally to this work.
† Deceased.
* vperiago@mundosano.org

**Data Availability Statement:** All relevant data are within the manuscript or supplementary files.

## Abstract

### Background

Chagas Disease (ChD) is a Neglected Tropical Disease (NTD) affecting 6 to 7 million people worldwide, mostly from Latin America. In Argentina, a national control program has been implemented since 1962, yet there are still an estimated 1.6 million infected individuals. Control programs were based almost exclusively on entomological surveillance and chemical control of households and were not continuous given a lack of coordination and resources. Argentina´s ChD program was originally vertical and centralized; later, it was partially and, in general, unsuccessfully transferred to the provinces. Herein, we describe the implementation of a control program for ChD with an ecohealth approach in rural settlements around the city of Añatuya, Santiago del Estero.

### Methods

The program included yearly household visits for entomological surveillance and control, health promotion workshops, and structural house improvements. Improved structures included internal and external walls and roofs, as well as the construction of water wells and latrines, and the organization and improvement of peri-domestic structures. Activities were carried out by specifically trained personnel except for house improvements, which were performed by the community, under technical guidance and provision of materials. Data was collected using standardized questionnaires for household characterization, entomological infestation status and chemical control activities.

### Results

This program was continuously implemented since 2005 with high community participation and adherence, incorporating 13 settlements and 502 households. During the surveillance

**Funding:** This study was initially funded (2005 to 2009) by a grant from Argentina´s Secretary of Science and Technology (Secretaría de Ciencia y Tecnología Argentina); MCA. Additional support was received from SC Jhonson and Chemotecnica, as well as ongoing support from Fundación Mundo Sano. The funders had no role in study design, data collection and analysis, decision to publish, or preparation of the manuscript.

**Competing interests:** The authors declare no competing interests.

phase, 4,193 domiciliary inspections were performed, and both the intra- and peri-domestic infestation rate were reduced from 17.9% to 0.2% (P < 0.01) and from 20.4% to 3%, respectively. Additionally, 399 households were structurally improved.

## Conclusion

The program is still ongoing and, after 14 years of implementation, has built social networks and collaboration between implementers and beneficiaries with a reduction of *T. infestans* infestation in the intra- and peri-domicile. This reduction, especially inside the household, has enabled access to diagnosis and treatment of the population, with minimal risk of re-infection.

### Author summary

Chagas disease (ChD) is caused by the parasite *Trypanosoma cruzi*, transmitted mainly through triatomine bugs of the Reduviidae family, genus *Triatoma*. These vectors are found in the Americas and *T. infestans* is the most common species in Southern Cone countries; closely tied to rural population that live in households with cracked walls and thatched roofs that serve as refuge for the bugs. Even though ChD can also be transmitted congenitally, through blood transfusions, or organ transplantation, among others, the main route of transmission is vectorial and programs for the prevention of this disease have traditionally focused on vector control through insecticide spraying. Argentina is the country with the greatest number ChD cases, and a national program of its control has been implemented since 1962, yet, due to a lack of sustainability, there are still 1.6 million infected individuals. Herein we describe a program that has been implemented since 2005 in rural areas from an endemic province of Argentina, Santiago del Estero, which worked together with the community, using an ecohealth approach, to monitor and control the vector and to improve the houses of the population to avoid the presence of the vector inside and prevent infection. The program achieved a reduction in triatomine infestation, especially in the domicile, therefore we hope it serves as an example to guide public health policy and enable diagnosis and treatment of ChD in endemic areas.

## Background

Chagas Disease (ChD) is a Neglected Tropical Disease (NTD) that affects around 6 to 7 million people worldwide: mostly individuals from Latin America [1]. In the early 1990´s, the World Health Organization (WHO) and its regional office in the Americas, the Pan American Health Organization (PAHO), coordinated disease control campaigns in different areas in cooperation with national authorities, contributing to the interruption of transmission in Brazil, Chile and Uruguay, which have been declared free of transmission due to the main vector of the disease in this area, *Triatoma infestans* [2]. Given the presence of different main vectors and transmission patterns depending on the region, several regional initiatives were created throughout the years since 1991, corresponding to the Southern Cone, Central America and Mexico, Andean countries, and Amazon countries [1].

Argentina is endemic for ChD and is estimated to be the country of Latin America with the largest number of people infected by *Trypanosoma cruzi*, approximately 1.6 million individuals

[3]. The main vector in Argentina is *T. infestans* which is present in most of the northern part of the territory [4] and although blood banks are controlled, congenital Chagas is a reality in endemic and non-endemic areas [5]. Control programs for ChD were originally focused on vector transmission using insecticides [6,7], which was the case of the National Chagas Control Program (Programa Nacional de Lucha contra la Enfermedad Chagas-Mazza) of the Ministry of Health of Argentina, started in 1961 [8] and formalized in 1962 [9] as a centralized and vertical program. Currently, Argentina is part of the Southern Cone Initiative to Control/Eliminate Chagas Disease (INCOSUR) and has already certified the interruption of vector transmission of *T. cruzi* by *T. infestans* in nine provinces between 2001 and 2018 [10].

In 1980, Argentinian Law 22.360 [11] dictated that each of the independent jurisdictions should implement the guidelines and regulation of the program; regulatory provisions for this law were decreed in 1982 [12]; this decentralization of ChD programs was occurring in other countries of the region as well [13,14]. Finally, in 2006 and 2007, laws 26.279 [15] and 26.281 [16] were passed, which included the mandatory control of ChD during pregnancy in the entire territory as well testing of all children and newborns. Despite these public health measures, control of ChD in Argentina, according to the milestones set by the 2021–2030 WHO roadmap, has yet to be achieved due to inconsistency in the application of these measures, a problem that is also present in other countries [17]. The recent regulatory provisions of Law 26.281 in Argentina, decreed in April 2022 [18], should help implement comprehensive control actions of surveillance, control, diagnosis, and treatment of ChD in the entire territory.

In this manuscript, the expansion of an entomological surveillance and control (S&C) program that started in 2002 in the city of Añatuya, Santiago del Estero, is described [19–21], including the design of a project with an ecohealth approach that was implemented in rural areas close to the city in 2005. Fundación Mundo Sano (FMS) set up its local office in Añauya, with a focus on ChD, in 2002 and it is still present to this day. This project started as a collaboration between different public and private organizations, led by FMS, including the National Ministry of Health, the specific area of Vector-borne Diseases (Dirección Nacional de Enfermedades Transmisibles por Vectores), the Ministry of Health from Santiago del Estero, the Municipality of Añatuya, Bunge and Born Foundation, CARITAS, and the Argentine Chamber of Plague Control (Cámara Argentina de Controladores de Plagas). From the beginning, the project promoted the inclusion and participation of community leaders, through training and capacity building, to create social networks for entomological surveillance in the intervention areas [19].

Santiago del Estero is located within the Gran Chaco Region [22], which is considered a hot spot for NTDs, including ChD [23]. This area has traditionally been an endemic area with the presence of the vector *T. infestans* and *T. cruzi* infected individuals [6,7,21] due to different environmental, social, and anthropological factors, which favor the presence of the vector inside the household [24]. More specifically, Santiago del Estero is in the *Monte* ecoregion of the Gran Chaco; an area originally composed of xerophytic forests, open woodlands, scrubs, savannas, and grasslands [25–28]. This ecoregion served as the original natural habitat for the sylvatic cycle of *T. infestans* where different native human populations lived.

Towards the end of the nineteenth century, a process of deforestation and the expansion of the railways brought new settlers into the region who started building their own houses as mobile laborers [25,29]. Eventually, as the railway was removed and the deforestation advanced, the settlers became more stable in the region forming different settlements throughout the area. The characteristics of the houses built by these settlers, the typical *rancho santiagueño*, and their livelihood based on family rearing of livestock, provided triatomine bugs another food source when the local fauna dwindled [22,29,30]. Unfortunately, there are very few published studies on the infection of this population through time, to accurately show the

evolution of ChD in the area. Nonetheless, data from very few localities of Santiago del Estero, have shown a decrease in human seroprevalence through time, especially in younger children [19,31–33] and a decrease of infection in triatomine bugs [7,34].

The National Chagas Disease Program has performed reiterated control actions with insecticides in the area since 1962 [19], which were later taken over by the Provincial Program under the Ministry of Health of Santiago del Estero, with the advantages and disadvantages of this decentralization [12]. Despite these programs, the interruption of vector transmission of *T. cruzi* was not achieved due to operational problems, discontinuity of actions, as well as the presence of conditions, both environmental and cultural, that favored the presence of the vector in the household and surrounding peri-domicile. Nonetheless, advances have been made in the southern departments of the province and in 2012, four of these were certified as free of transmission by *T. infestans* [35].

The original S&C program that was circumscribed to urban areas of Añatuya in 2002, was expanded to rural areas in 2005, with the incorporation of structural modifications of the *rancho*, through community participation, training, and social networking. House improvements were added not only to decrease the risk of re-infestation by kissing bugs, but also to improve the quality of life of the inhabitants in this area [20]; who live in suboptimal conditions with high vulnerability. The improvement of households is currently in line with the guidelines of the PAHO/WHO for control of vector-transmitted diseases [36], improved housing conditions [37,38] and in alignment with the United Nations Sustainable Development Goals (UN-SDGs) [39], taking into consideration hygiene, water, and sanitation (WASH).

In this manuscript, we describe the different components of the approach used throughout the study, based on an ecological perspective, or ecohealth approach, which tries to consider not only the disease, but also the interrelation between the physical, social, and cultural aspects of health; taking into consideration that behavior change requires interventions at different levels [40]. We also report the evolution of the program, which is ongoing, after 14 years of implementation (2005–2019), with respect to its growth, community participation, entomological indices, challenges, and lessons learned. The sustained work through these years, through a public-private partnership, has achieved a decrease in intra-domiciliary infestation by triatomine vectors, permitting access to specific diagnosis and treatment for ChD, with minimal risk of re-infection, in these rural communities from Santiago del Estero.

## Methods

### Study area and program design

The data herein is from an ongoing entomological S&C program implemented in rural areas of the Departments of General Taboada and Juan F. Ibarra, Santiago del Estero Province, Argentina (28˚27'32.64" S, 62˚50'15.24" O). Añatuya is the main city of the Department of General Taboada; it is the commercial and administrative center of the region, and its population is approximately 38,000 inhabitants that live in 9,073 houses [41]. The rural S&C program started in 2005 with a grant from the National Secretary of Science and Technology (Programa Federal de Innovación Productiva—PFIP 2004/1), in agreement with the Ministry of Health and Social Development of the Province of Santiago del Estero and the collaboration of the National Agricultural Technology Institute (INTA). At that time, selection of settlements to be included, were based on different characteristics: distance of the settlement from the local FMS office in Añatuya, total number of households, whether FMS had worked previously in the area, community build up characteristics and sanitary conditions. The different components of the ecohealth approach used in this study included gathering physical and demographic data, giving informational, educational, and training (IET) workshops on ChD, chemical

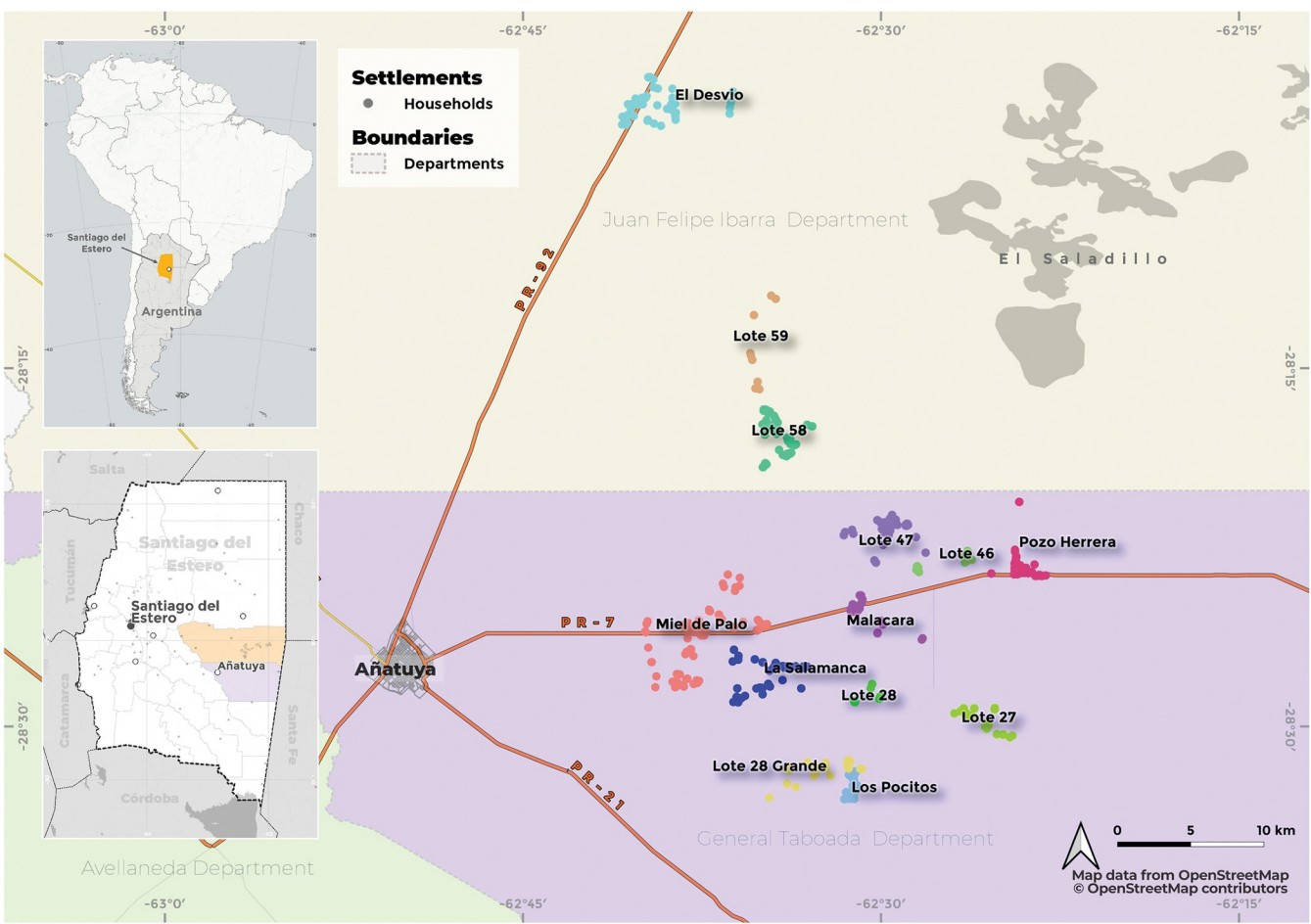

**Fig 1. Study area: rural settlements under entomological surveillance and control in the Departments of General Taboada (shaded in purple) and Juan F. Ibarra (shaded in grey), Santiago del Estero, Argentina.** The households within each settlement are represented in different colored points (El Desvío, Lote 58, Lote 59, La Salamanca, Lote 27, Lote 28, Lote 28 Grande, Lote 46, Lote 47, Los Pocitos, Malacara, Miel de Palo, and Pozo Herrera). Map data available at www.openstreetmap.org/export, map data license at www.openstreetmap.org/copyright. Map created with QGIS version 3.26.1-Buenos Aires.

control of the entire settlement prior to surveillance implementation, promotion of social networks [19], implementation of surveillance, and implementation of sanitary house improvements.

The temporal period analyzed in this study is from June 2005 to December 2019. The area under influence of the ongoing program comprises 13 rural settlements with 526 households, 1,720 individuals and located less than 50 km from the City of Añatuya: El Desvío, Lote 58 and Lote 59 (all three located in Juan F. Ibarra Department), and La Salamanca, Lote 27, Lote 28, Lote 28 Grande, Lote 46, Lote 47, Los Pocitos, Malacara, Miel de Palo, and Pozo Herrera, located in General Taboada Department (Fig 1).

## Surveillance and control

Every time a new settlement was included in the S&C program, an initial entomological evaluation was performed to estimate the baseline infestation of *T. infestans*. Regardless of the outcome of this initial inspection, chemical control was subsequently performed in all the households, both in the intra- and peri-domicile [42,43]. During this visit, oral and written consent was obtained from a responsible adult to enter the household and collect the different

types of data; oral consent was also obtained prior to each subsequent intervention. Through standardized questionnaires (S1 File) and visual inspection, sociodemographic, entomological infestation status (presence of *T. infestans* and/or other triatomine bugs), household characteristics and epidemiological data of each family were recorded. Each house was georeferenced (eTrex Legend HCx, Garmin Ltd., USA), given an identification number (ID) and pictures of the household and peri-domicile were taken. Information and graphic material related to ChD was also shared.

After the initial visit, a S&C system was established, including households and other buildings that might be present in the settlement, such as schools or health posts. During the initial visit and each subsequent one, the field agents recorded the status of the building: inspected (if the home was indeed receptive to inspection), closed (if the house could not be inspected at the time of the visit due to lack of inhabitant presence or a responsible adult for consent), reticent (if the family did not want their home inspected), uninhabited (if the house was closed and vacant), or dismantled (if the building no longer existed).

Entomological inspections were conducted by the hour/man method [19,42], where two specifically trained agents inspected each house using an irritant spray ICONA-Espacial (Tetramethrin 0.2%) to dislodge any bugs from their refuge. The hour/man method consists in systematically searching the interior (intra-domicile) and exterior (peridomicile) of a house by one agent during an hour to determine the presence of triatomine bugs. If there are two agents performing the inspection, then the time is reduced by half. All the intra-domicile and peri-domestic structures (henhouses, storerooms, kennels, animal pens, toilets, or latrines, etc.) were included in the inspection, following guidelines [19,42] and recommendations from the National Ministry of Health [43].

The result of the inspection, either absence of presence of triatomine bugs, was recorded in an inspection sheet (S2 File) as negative or positive. If at least one triatomine bug was found, either in the intra- or peri-domicile, or both, a house was considered positive. Details on the place of discovery of the bug (in the intra- or peri-domicile and in what specific structure or place), the life stage of the specimen found (egg, nymph or adult), and semi-quantitative number of bugs found (1–1 to 10 specimens; 2–11 to 50 specimens; 3–50 to 100 specimens; and 4—more than 100 specimens), was recorded in paper form. The presence of triatomine eggs or exuviae (exoskeleton after molting), and the trace of characteristic defecation stains of triatomines were recorded. If no adult or nymph *T. infestans* specimens were observed, the house was still considered negative.

The inspection process was always performed in the same manner and there are standardized operating procedures (SOPs) to assure that all agents are specifically trained and begin in the intra-domicile followed by the peri-domicile. After the inspection was finalized, all the recorded information was shared with the person who accepted the inspection and a copy was given, together with specific information on prevention, so that any doubts could be cleared. Each of the houses within the program was inspected at least once every 12 months and the completion of the inspection in each settlement is referred to as a cycle or round.

In subsequent surveillance of the settlement under the program, the overall results were analyzed to determine the need for control. Houses with the following characteristics were controlled by insecticide spraying both in the intra and peri-domicile during each cycle of inspection: (a) positive houses, (b) negative houses located in a radius of less than 500 meters from a positive house and, (c) other structures that have certain epidemiological, demographic or building characteristics that make them highly suitable for triatomine refuge. For example, houses that store wood for fire or that work with coal and wood production, houses with the presence of other triatomine species other than *T. infestans*, and houses of other family members that have close ties to the positive house.

All the insecticide application information was entered in a registration form specific for *T. infestans* control actions as described in S3 File. Siperthrin (Beta-Cypermethrin 5%—Asimethrin, Chemotecnica S.A., Buenos Aires, Argentina), a pyrethroid insecticide of flowable formulation, was used, following the dosage (50 mg a.i./m2) from guidelines [42] and recommendations [43].

### Sanitary improvement of households

The traditional housing from the area is the *rancho*, as previously described [19,21,44], which is composed of a large peri-domestic backyard consisting of different items no longer in use; wood and adobe brick constructions which are used as a kitchen and animal pens (mostly for goats and chickens) which serve as ideal refuge for *T. infestan*s. The different households in the area are arranged in a dispersed manner and may be separated by distances ranging from meters up to several kilometers.

The sanitary improvement of households seeks to improve housing in rural areas to diminish the risk of triatomine re-infestation and to support access to necessary basic needs to be able to adopt healthy habits; not including improvement of other buildings like schools or health posts. Usually, once the settlement was under S&C and a bond had been established with the community, the possibility of house improvement was offered. If the community agreed to adhere to the building aspects of the program, leaders of the community were identified to act as coordinators. Each of the leaders represented a small group of families and acted as the point of contact to conform social networks between the program and the families [19,42,45]. These improvements were usually conducted one settlement at a time.

As a first step to implement household improvement in a settlement, a meeting was held with the entire community to explain the program itself. The improvements in the houses were carried out by members of the community themselves under the leadership and supervision of a master builder. Therefore, the program was performed as a community project, where everyone collaborated with the construction of everyone's houses, including houses of families that could not collaborate in their own improvements due to age, disability, health issues or other situations. Working groups of approximately 10 to 15 families were created depending on the proximity of the houses.

The project's master builder visited all the households in the settlement, and based on the previous characterization of each house, recorded the different improvements individually needed to calculate the amount of material required for all the households and to be able to organize the work. The improvements for each household were defined based on specific characteristics of each house, defining structural needs and health-related improvements together with the family and the community. Materials were delivered to a single house or to more than one house within the settlement, depending on the distance, and then each family transported the materials for their own use. As the improvements were performed, the master builder kept track of advancements on a spreadsheet that was then digitized for monitoring purposes.

Before household improvements were initiated, IET community workshops on ChD were held, emphasizing the habits, or building issues that may favor the presence of triatomines in the household. Other workshops given included topics such as the improvement of the entire household with traditional materials, transferring of technology to improve animal pens, chicken coops and other structures; talks on the provision and storage of safe water, training on the construction of a water well; talks on the correct handling of human excreta and training for the construction and maintenance of improved latrines; among others.

After the IET workshops, demonstrative practice workshops were held; the master builder chose a home according to the family´s ability to make the improvement themselves, and

demonstrated how to make each improvement. The demonstration workshops and the improvements of all the houses were carried out in stages, i.e., the second improvement did not begin until the first one was finished in all the houses in the settlement and so on. The program provided all the materials to carry out the improvements free of charge. The construction elements chosen were those that could be easily obtained in the area, prioritizing autochthonous materials, such as dirt, the sap from a local plant (*Elionurus muticus*), wood and sticks, to facilitate maintenance through time.

The household improvements included in the program are listed below, in the order they were performed:

- Construction of a well for water storage (capacity of 3,000 liters);

- Construction of an outhouse with a toilet and sink for hand washing;

- Construction of a space inside the home for adequate handling and processing of food;

- Plastering and whitewashing of all inside walls;

- Improvement and waterproofing of all roofs;

- Plastering and painting of all external walls;

- Household and peri-domicile cleaning and environmental management (including animal pens, chicken coops, stoves, storerooms, barns, etc.) and construction of animal pens. These are relocated if needed so that they are at least 50 meters from the household. These activities are transversal to the rest of the improvements; they begin with the water well construction and continues as the program progresses.

At the end of the construction of water wells and latrines, an informative workshop on maintenance and hygiene of both structures was provided. Through the years, not all the homes received the same type and quality of improvements since, as the program progressed, better forms of construction were incorporated. These were based both on the acquired experience and people's needs. In the beginning (2005), improvements included working on the animal pens, building of a water well and a simple latrine. Starting in 2010, the water well began to be built with a lid and its capacity was increased. The latrine was also improved, making the space larger, with a roof, adding a cesspool and using a factory toilet. In 2016, the construction of a sink for handwashing was added in the bathroom and external cement plaster was incorporated. Finally, in 2017, a clean area appropriate for food preparation was added as an improvement within the household. The interior of the houses, roof improvement and waterproofing have remained the same since the beginning.

## Statistical analysis

Data was collected in the field using standardized forms as per SOP and then entered in the field office into a customized Geographic Information System (GIS), a data management platform that was specifically designed and developed for the program as previously described [21]. This platform includes records of all the household locations (georeferenced), their sociodemographic information and the results of each inspection, including the entomological data, as collected in the different paper forms (S1–S3 Files).

The different entomological indices were calculated as follows: % infestation = (number of infested houses/ number of inspected houses) x 100; % intra-domiciliary infestation (IDI) = (number of houses infested in the intra-domicile/ number of inspected houses) x 100; % peri-domiciliary infestation (PDI) = (number houses with infestation in the peri-domicile/ number of inspected houses) x 100; and %IDI & PDI = % IDI = (number houses positive in both the

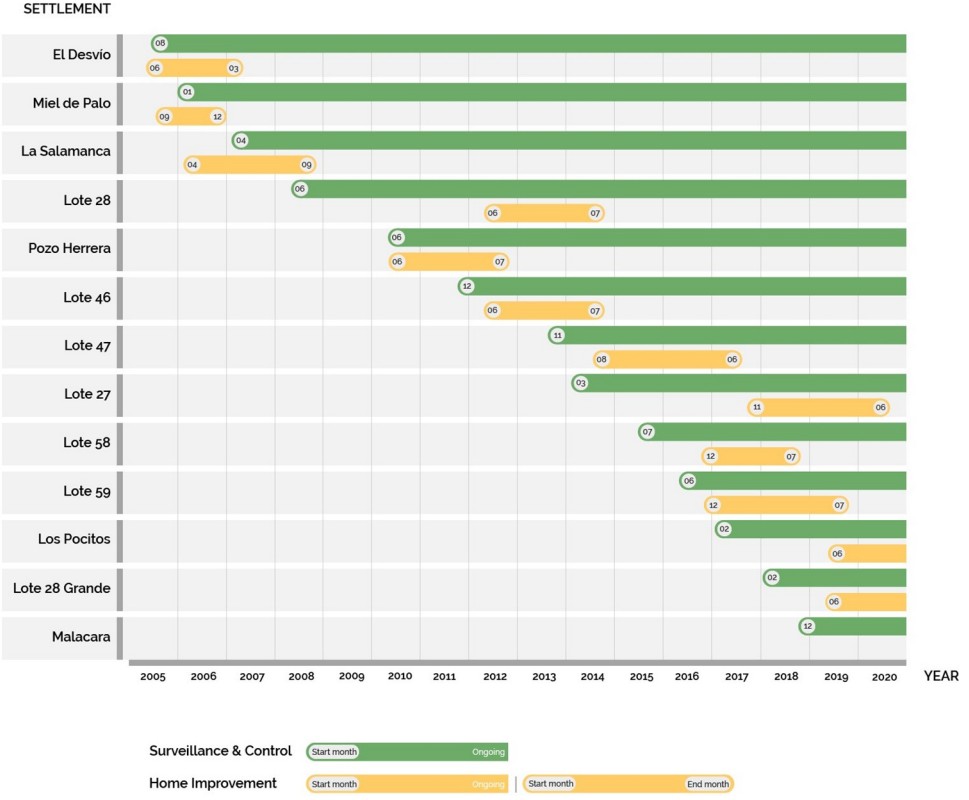

**Fig 2. Timeline and evolution of the surveillance and control program implemented in the rural settlements from the Departments of General Taboada and Juan F. Ibarra in Santiago del Estero (Argentina).** The different settlements included are listed by the year they were incorporated, showing their incorporation in the surveillance and control activities (in green) as well as the home improvement activities (in yellow).

intra-domicile and the peri-domicile/ number of inspected houses) x 100. Statistical analyses were performed using R-studio (v. 4.1.3). The N-1 Chi-square test was used to compare proportions of infestation with a level of statistical significance of $P < 0.05$. For evaluation of paired households, either the McNemar or the binomial exact test was used.

## Results

The program started in 2005 in El Desvío, Miel de Palo, La Salamanca, Lote 28, and Pozo Herrera, rural settlements located close to Añatuya. The funding at the time was for three years and activities started in El Desvío. As the program grew, selection of the settlements to be included through time followed the initial criteria. A timeline of the program, with its different components, is summarized in Fig 2. Nonetheless, as the neighboring settlements became aware of the program, some asked FMS to be included, usually through written letters (S1 Fig).

### Surveillance and control

All the incorporated settlements are currently still under S&C and herein we report the data from 2005 to 2019. During the study period, 4,193 domiciliary visits were conducted comprising a total number of 13 settlements and 502 households. During S&C, some settlements were visited more than once a year, this is why the number of inspected houses might be higher than the number of households under the program (Table 1).

**Table 1. Summary of the chronological evolution of the surveillance and control program implemented in the rural settlements from the Department of General Taboada and Juan F.** Ibarra (Santiago del Estero) Argentina from 2005 to 2019. The table details the number of houses included in the surveillance and control stage.

| Year | Name of settlements included | Receptive no. of households/total No. visited | % participation | Cumulative no. of settlements included | Current no. of completed S&C cycles* |
|------|------------------------------|-----------------------------------------------|-----------------|----------------------------------------|--------------------------------------|
| 2005 | El Desvío | 39/43 | 90.7 | 1 | 17 |
| 2006 | Miel de Palo | 211/211 | 100 | 2 | 18 |
| 2007 | La Salamanca | 202/202 | 100 | 3 | 21 |
| 2008 | Lote 28 | 147/147 | 100 | 4 | 14 |
| 2009 | None | 92/92 | 100 | 4 | 14 |
| 2010 | Pozo Herrera | 252/259 | 97.3 | 5 | 12 |
| 2011 | Lote 46 | 247/256 | 96.5 | 6 | 9 |
| 2012 | None | 199/209 | 95.2 | 6 | 9 |
| 2013 | Lote 47 | 280/289 | 96.9 | 7 | 7 |
| 2014 | Lote 27 | 482/501 | 96.2 | 8 | 8 |
| 2015 | Lote 58 | 413/440 | 93.9 | 9 | 5 |
| 2016 | Lote 59 | 546/589 | 92.7 | 10 | 4 |
| 2017 | Los Pocitos | 408/426 | 95.8 | 11 | 4 |
| 2018 | Lote 28 Grande/ Malacara | 240/250 | 96.0 | 13 | 2/1 |
| 2019 | None | 435/464 | 93.8 | 13 | NA |

NA: Non-Applicable. *Each of the houses within the program was inspected at least once every 12 months and the completion of the inspection in each settlement is referred to as a cycle or round. The current number of cycles refers to the number of completed cycles as of the year 2019.

During the different cycles of surveillance, and depending on the incorporation of new settlements, the entomological indices varied as the S&C program progressed (Fig 3 and S1 Table). From 2005 to 2019, the overall evolution of infestation curves shows a general decrease as the program progressed; with a significant decrease in total infestation from 46.2% to 5.1%

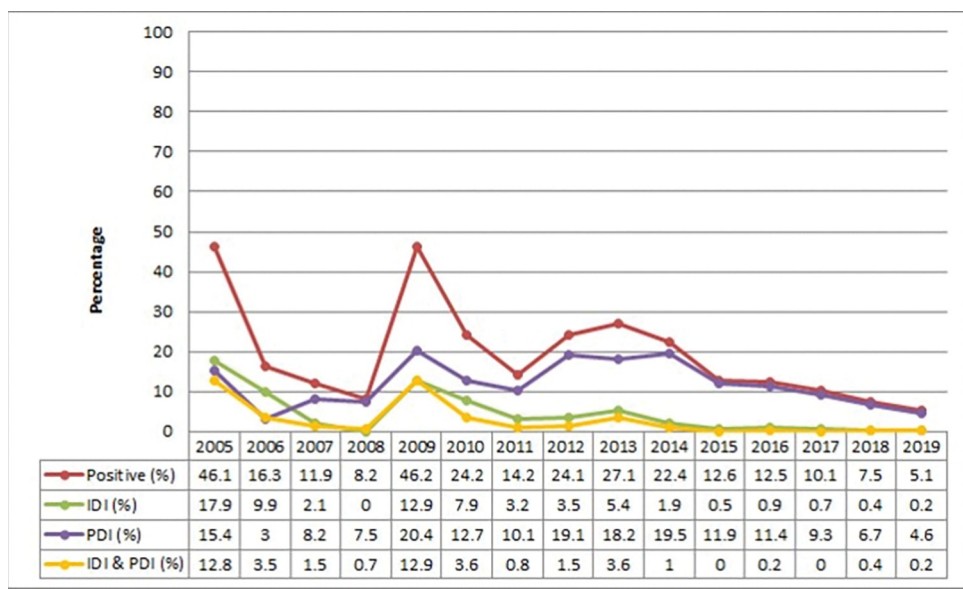

| | 2005 | 2006 | 2007 | 2008 | 2009 | 2010 | 2011 | 2012 | 2013 | 2014 | 2015 | 2016 | 2017 | 2018 | 2019 |
|---|---|---|---|---|---|---|---|---|---|---|---|---|---|---|---|
| Positive (%) | 46.1 | 16.3 | 11.9 | 8.2 | 46.2 | 24.2 | 14.2 | 24.1 | 27.1 | 22.4 | 12.6 | 12.5 | 10.1 | 7.5 | 5.1 |
| IDI (%) | 17.9 | 9.9 | 2.1 | 0 | 12.9 | 7.9 | 3.2 | 3.5 | 5.4 | 1.9 | 0.5 | 0.9 | 0.7 | 0.4 | 0.2 |
| PDI (%) | 15.4 | 3 | 8.2 | 7.5 | 20.4 | 12.7 | 10.1 | 19.1 | 18.2 | 19.5 | 11.9 | 11.4 | 9.3 | 6.7 | 4.6 |
| IDI & PDI (%) | 12.8 | 3.5 | 1.5 | 0.7 | 12.9 | 3.6 | 0.8 | 1.5 | 3.6 | 1 | 0 | 0.2 | 0 | 0.4 | 0.2 |

**Fig 3. Percentage of households with the presence of triatomine bugs (infestation) per year are denoted in red.** The green line represents the percentage of households with intra-domicile infestation (IDI) of triatomine bugs, the purple line represents the percentage of households with peri-domicile infestation (PDI) and the yellow line represents the infestation of both in the intra- and peri-domicile (IDI & PDI) for each of the surveillance and control rounds.

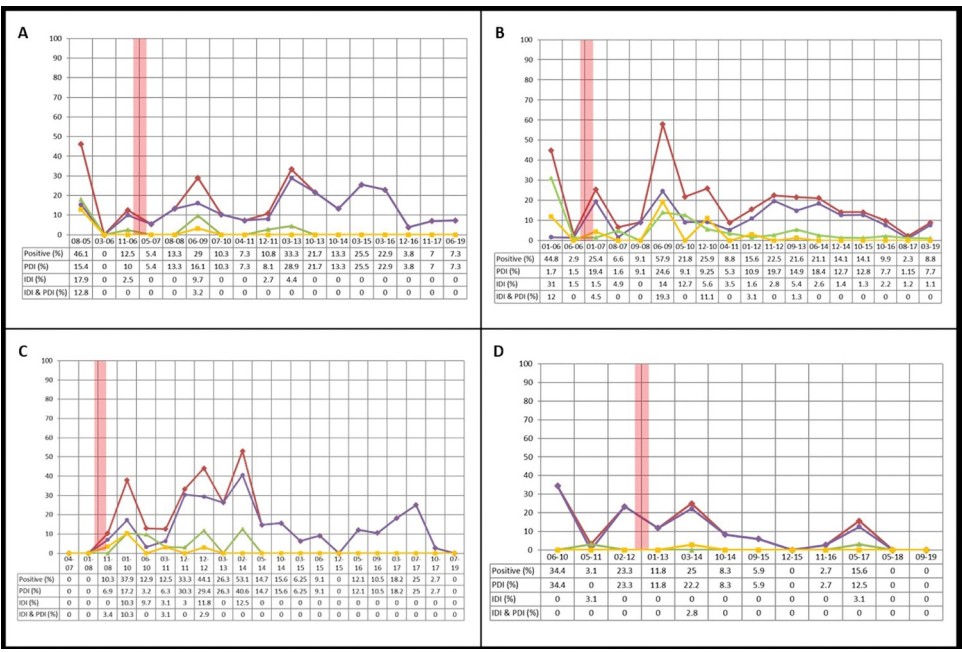

**Fig 4. Percentage of households with the presence of triatomine bugs (infestation) per surveillance and control cycle are denoted in red.** The green line represents the percentage of households with intra-domicile infestation (IDI) of triatomine bugs, the purple line represents the percentage of households with peri-domicile infestation (PDI) and the yellow line represents the infestation of both in the intra- and peri-domicile (IDI & PDI) for each of the surveillance and control rounds. The vertical wide red lines indicate the end of house improvement in each of the settlements. **A.** El Desvío with 17 completed cycles, **B.** Miel de Palo with 18 completed cycles, **C.** La Salamanca with 21 completed cycles and **D.** Pozo Herrera with 12 completed cycles.

(P < 0.01). The peri-domicile infestation (PDI) rate fluctuated throughout the study period and ranged between 3% to 20.4%, while the intra-domiciliary infestation (IDI) in decreased significantly, from 17.9% to 0.2% (P < 0.01). Moreover, a focus on the overall positivity of households which remained constant from 2010 to 2019 in the settlements of El Desvío, Miel de Palo, La Salamanca, Lote 28, and Pozo Herrera, shows that there was a significant decrease in infestation (P < 0.01) using only those houses that were present both in 2010 and 2019 using a paired test (S2 Table).

Depending on the settlement, different infestation patterns were observed. For example, El Desvío, located 44 km from Añatuya, on provincial route (PR) 92, has already gone through 17 complete surveillance cycles. As shown in Fig 4A, the IDI decreased from 23.1 to 0% (P < 0.01), while the PDI fluctuated despite controlling with insecticide during 12 cycles (S2 Table). The data from Miel de Palo, the largest settlement in the study area, also shows a significant decrease of IDI over time (from 29.4 to 1.1%, P <0.01), while again the PDI fluctuated (Fig 4B) even though insecticide control was performed during 13 cycles. This same pattern was observed for the settlements subsequently incorporated: La Salamanca (Fig 4C) with 21 completed cycles, Lote 27, Lote 46, Lote 47, Lote 58, Lote 59 and Malacara. Nonetheless, in the last cycle, during 2019, of La Salamanca and Malacara, there was cero infestation in both the intra- and peri-domicile.

On the other hand, in Pozo Herrera, located on both sides of PR-7, the farthest east from all the sites under the program, with 12 completed cycles, the IDI has never been high compared with the PDI (Fig 4D), leveling at 3.1% at its highest point and decreasing to 0% since 2018 (P<0.05). As in the other settlements, the PDI fluctuated through the cycles, but with significant reduction; from 34.4% to 0% (P < 0.01), remaining at zero in all the households during

**Table 2. Summary of the evolution of the sanitary house improvement component of the surveillance and control program implemented in the rural settlements from the Department of General Taboada and Juan F.** Ibarra (Santiago del Estero) Argentina from 2005 to 2019. The table details the number of houses included in the surveillance and control stage, including the name of the settlement, the current number of houses in each settlement and the last column refers to the cumulative number of houses improved as the program progressed.

| Year | Settlement | Current No. of houses per settlement | Cumulative No. of houses improved |
|------|-----------|-------------------------------------|-----------------------------------|
| 2005 | El Desvío | 62 | 41 |
| 2006 | Miel de Palo | 107 | 116 |
| 2007 | La Salamanca | 41 | 149 |
| 2008 | Lote 28 | 14 | 149 |
| 2010 | Pozo Herrera | 42 | 182 |
| 2011 | Lote 46 | 16 | 182 |
| 2012 | NA | NA | 198 |
| 2013 | Lote 47 | 57 | 198 |
| 2014 | Lote 27 | 26 | 253 |
| 2015 | Lote 58 | 54 | 253 |
| 2016 | Lote 59 | 9 | 315 |
| 2017 | Los Pocitos | 34 | 339 |
| 2018 | Lote 28 Grande/ Malacara | 27/38 | 339 |
| 2019 | NA | NA | 399 |

NA: Non-applicable

the last two cycles. A significant reduction in PDI was also observed in Lote 28, Lote 28 Grande and Los Pocitos, reaching zero infestation during the last cycle (2019). A summary of the entomological indexes both by year and by settlement is found in S1 and S2 Tables.

In all the settlements included in the program, control actions through insecticide spraying both the intra- and peridomicile were performed every time a house was positive, regardless of the location of the bugs (intra- or peri-domicile). The number of S&C cycles also varies depending on the percentage of re-infestation found; settlements with high re-infestation rates were visited and chemically controlled, more often (El Desvío, Miel de Palo, La Salamanca, Lote 27, Lote 47, Lote 58 and Lote 59) (S2 Table). The specific places were the triatomine bugs were found, both in the intra- or peridomicile, or both, where recorded since 2010. The total amount of places were bugs found from 2010 to 2019 was 470; 81.5% (n = 383) in the peridomicile which was significantly higher than the 13.0% (n = 61) found in the intra-domicile (P < 0.01) and 5.5% (n = 26), in the intra- and peridomicile. When analyzing these specific places, in the peridomicile, a significantly higher proportion of triatomine bugs were found in animal pens (35.8%, n = 137) compared to chicken coops (21.7%, n = 83; P < 0.01) and 13.1% (n = 50) were found in storage rooms. In the intra-domicile, more than half of the bugs were found in the sleeping quarters (68.9%, n = 42); this was significantly higher than in the rest of indoor sites (P < 0.01).

## Sanitary house improvement

After the settlements were under S&C, house improvement was offered to the community and organized as previously described. It is important to note that the number of households under the program varies through time, since inhabitants build new houses, take down old ones or move from one settlement to another or from a settlement to the City of Añatuya or elsewhere. The first settlement to start house improvements was El Desvío with the improvement of 36 homes in 2005. Table 2 details the settlements that were incorporated for

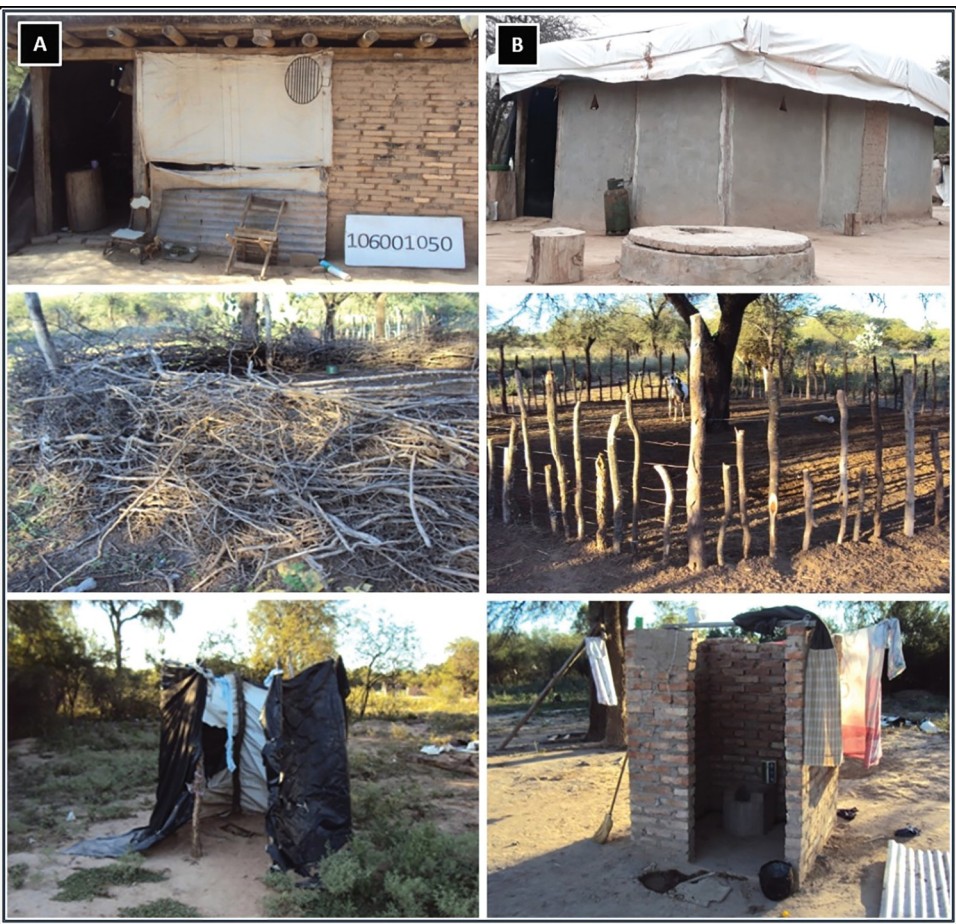

**Fig 5. Sanitary house improvement in a settlement from Añatuya, Santiago del Estero, Argentina under the surveillance and control program implemented from 2005 to 2019).** This is an example of the improvements made, showing the different improvements before (column A) and after (column B). The pictures include a household with and without a well, the peri-domicile with an animal pen, and a latrine.

improvement each year, showing the year of incorporation and the cumulative number of households improved as the program progressed. Since this program has been implemented for many years, the current number of houses in the settlements has changed throughout the years and is also reflected in the table. Fig 5 shows an example of the house improvements made in each of the settlements.

## Discussion

In this study, the evolution of a S&C program for ChD with an ecohealth approach, implemented in rural areas of Añatuya, Santiago del Estero Argentina, is described. Since its beginning in 2005, it grew from the inclusion of two settlements to 13 settlements in 2019. Currently, the program is ongoing, and a new settlement was included in 2022. From the beginning, the program was comprehensively designed with the aim to decrease vector transmission of *T. cruzi* through both S&C and the improvement of sanitary conditions of the inhabitants through amelioration of the structural conditions of their living spaces. For this reason, the ecohealth nature of the program focused on the building of social networks and collaboration between the inhabitants and the program executors to ensure adherence and

sustainability, with the transfer of skills aimed at empowerment of the community beyond the scope of the specific ChD-related project, thus contributing to the improvement of the living conditions of the beneficiaries [19].

Traditionally, ChD prevention and control was based on spraying of households with insecticide and screening of blood donors [4,5], with few programs tackling the house improvement component of the problem. One of the first countries to pioneer these types of interventions and highlight the importance of the household was Venezuela [46]. Subsequently, other countries began including house improvement as part of their control programs [14,47] or as pilot experiences in specific areas [48–57], with varied results and difficulties in sustainability. Through these different experiences, as well as extensive research, the factors of the rural household that contribute to the presence of kissing bugs have been determined [7,49,50,52,53,58–61]. For example, wall plastering, roof waterproofing, organization of the peridomicile, improvement of peridomestic structures, including animal pens, and limiting the entrance of chickens and other animals in the house.

In the Province of Santiago del Estero, given the lack of adequate housing in terms of quality and amount, specific housing laws were passed [62,63] with the objective to improve the sanitary conditions, health, and education of the population, as well as the construction of new houses, including the elimination of the more precarious rural *ranchos* [64] and moving disperse rural population to areas closer to main roads. This provincial program still exists and although it hasn't been evaluated, most houses did not reach the targeted dispersed rural populations. Moreover, in those cases where it did, observations show that some families use the houses for storage or other purposes while the original *rancho* remains intact [63]. Different studies highlight the importance of community participation [7,47,51,52,54,56] and the need for a comprehensive improvement of households, given that tackling only one of the factors that facilitate triatomine infestation has proven insufficient [7,48,49,58–61,65,66]. The experience described herein took into consideration the local traditions and way of life of the inhabitants, as well as their experiences and suggestions to together improve the situation of vector transmission of ChD in the area as well as other issues such as lack of water or basic sanitation. The perception of those affected by the problem being tackled needs to be considered, since prevention of ChD, or any other disease, may not be a goal for community members, while improving quality of life is [67].

Due to the use of an ecohealth approach in a public-private collaboration, the program is still ongoing after all these years, with continuous ChD S&C in 14 settlements. This expansion considered the original inclusion criteria of the project and incorporated some neighboring settlements that requested their inclusion. Moreover, the program has been successful in lowering the infestation rates in the area as evidenced by a reduction in the overall infestation rates in the settlements under the program from 2005 to 2019. The significant reduction of intra-domiciliary infestation from 17.9% in 2005 to 0.2% (P < 0.01) is of epidemiological importance given the feeding behavior of *T. infestans*. Despite these achievements, a limitation of the program is that it was formulated as a public health program to address the problem of ChD in the area, in collaboration with local authorities and organizations, with the goal of implementing a sustainable prevention and control program; therefore, non-intervened control settlements were not used for comparison.

The peri-domiciliary infestation (PDI) fluctuated more through the years decreasing from a peak of 20.4% in 2009 to 3% in 2019 (Fig 3). This fluctuation in the peri-domicile is to be expected, as observed in other studies [7,50,51,53,56,59,68] and is probably due to the presence of animals (including chickens and goats, but also dogs) and the passive transfer of triatomines from other areas, especially with firewood. In some settlements, including El Desvío, Miel de Palo, La Salamanca, Lote 27, Lote 46, Lote 47, Lote 58, Lote 59 and Malacara, the IDI was kept

low, but the PDI was more variable with the need for frequent chemical control activities (S2 Table).

During inspection, most of the triatomine bugs in the peri-domicile, were found in association with animal pens and chicken coops. While in other settlements, like Pozo Herrera, Lote 28, Lote 28 Grande, and Los Pocitos, IDI reached zero and the PDI also fluctuated but was significantly reduced through the years and eventually also reached zero. Additionally, the settlements themselves have gone through different demographic, structural and land changes; evidenced by an increase in the number of houses (S2 Table). For example, La Salamanca went through a re-localization of families during 2009–2010. The old houses had to be dismantled and a new area had to be cleared of vegetation for construction of new houses; a club for recreational activities was also constructed. This might be the cause of the fluctuation of entomological indexes observed during 2011–2014 (Fig 4C).

We believe that the ecohealth approach applied, taking into consideration individual and social determinants, stimulated the participation of the population, reflected through a participation greater than 90% in the S&C activities throughout the years, as observed in other intervention areas of Guatemala and Ecuador [49,51,52,55,56]. By providing improvements and listening to needs that went beyond the specific problem of ChD, individual and community engagement was achieved and sustained. The improvement of the households and the environmental management of the peri-domicile also facilitated surveillance activities and probably increased the effectiveness of the chemical control actions [7,52,57,69–71]. Thus, the continued and effective treatment with insecticides is probably acting as the mayor driver to lower the triatomine population, as observed in other studies conducted in rural villages from this province [72,73] and in other areas [57,70], given that only house improvement without insecticide control is not enough to drive down the triatomine population in the peridomicile of rural houses in an area where the sylvatic cycles is ongoing and the presence of domestica animals serves as a food source [57–61]. Therefore, the combined approach with house improvement and insecticide spraying showed to be advantageous, driving intra-domestic infestation significantly low, and controlling peri-domestic infestation more efficiently due to improvement of peri-domestic structures and environmental management for more efficient insecticide spraying.

With respect to the impact of the program on the infection rate of the population, domestic transmission is considered interrupted when less than 1% of children under the age of 5 years have positive serology [74]. Baseline data for the project is available from serological surveillance conducted during 2003 in rural settlements of General Taboada [19], with an overall *T. cruzi* infection prevalence of 5.2% in children from 6 months to 14 years of age (n = 581); 0 prevalence in children 6 months to 2 years (n = 21), 2.6% prevalence in children 3 to 5 years (n = 71), and 5.7% prevalence in children 6 to 14 years. FMS has been performing different serological screening activities in rural settlements during 2016 to 2019 with the local municipality, specifically in some of the same rural settlements included herein, analysis of this data will serve to show if the prevalence in children under the age of 5 years is less than 1%.

The availability of specific knowledge on ChD and accessible technological tools for house improvement has enabled these rural communities of Santiago del Estero to deal with, and overcome, the public health problem of ChD. This was possible due to insecticide control of the houses, but also due to the incorporation of construction, hygienic and organization practices compatible with the elimination of *T. infestans*. The access to affordable and available variations in construction technologies allowed these communities to adapt their houses to appropriate sanitary conditions, given that the *rancho* has characteristics that the locals consider unique and irreplaceable given its construction with materials that are locally available, adaptable to high temperatures and usually with a spacious porch. This respect for local needs

and traditions, together with community participation and involvement, use of local materials, and continued presence and guidance from local institutions and authorities assured sustainability of the program.

The program described herein, incorporated the collection of standardized data (SOP´s), adjusted the type of the data collected and incorporated a data platform with GIS. Given the impact of this intervention, with high acceptance by the population, we believe this experience and its records can be used for transference to other countries of the region, especially those that have their own similar characteristics and experiences [49,52,56,57,75]. One of the important aspects of the program is its flexibility, changing through time and adapting to different situations and difficulties encountered through the years, making it dynamic, with the integration of proposals suggested by the beneficiaries themselves. Nonetheless, there is a lot of information that has not yet been considered to explain the differences observed between settlements with respect to re-infestation. The specific characteristics of each settlement related to their customs or occupation, closeness to main roads, distance between houses in each settlement, size of the settlement, natural habitats surrounding each settlement and advancement of the agricultural border [21,24,76,77], with its implication on land tenure, are all factors that should help determine the differences observed.

## Conclusions

The program described within has shown that the ecohealth approach is effective for long term control of ChD. The evidence generated through this intervention, with continuous S&C, community participation, sanitary improvement of houses, and lowering of re-infestation indices, should serve to guide control programs and shape public policies for the control of this neglected disease. More specifically, when implementing a ChD control program at the local level, authorities should take into consideration the specific transmission pattern observed, the local customs and organization of the community, as well as the tools and resources available. This experience in rural areas of Santiago del Estero has shown that to effectively transfer a control program, assuring feasibility and sustainability, it´s important to work together with the inhabitants and local organizations to sum efforts and thus facilitate continuous implementation of the activities. The S&C activities described within have been implemented by FMS throughout all these years. In the future, we hope to transfer the surveillance component of the program to the community; we believe that the social network structure built throughout these years should serve a foundation for this transfer. The success of this program in rural areas of Santiago del Estero, with a reduction in the risk for vector transmission, should enable the population to get access to diagnosis and etiological treatment of ChD.

## Supporting information

**S1 File. Questionnaire used for household and socioeconomic characterization.**
(DOCX)

**S2 File. Worksheet used during the household inspections to record the presence of *Triatoma infestans*.**
(XLSX)

**S3 File. Worksheet used to record chemical control for *Triatoma infestans*.**
(DOCX)

**S1 Fig. An example of a letter received in April 2013 from community members of Lote 27, Departamento de Taboada, Santiago del Estero (Argentina), asking to be included in**

**Mundo Sano´s surveillance and control program.** The panel on the right contains the transcription of the letter in English.
(TIF)

**S1 Table. Summary of entomological indexes obtained during the different rounds of surveillance and control implemented from 2005 to 2019 in the rural settlements from the Departments of General Taboada and Juan F. Ibarra, Santiago del Estero, Argentina.**
(DOCX)

**S2 Table. Summary of entomological indexes by settlement obtained during the different rounds of surveillance and control implemented from 2005 to 2019 in the rural settlements from the Departments of General Taboada and Juan F. Ibarra, Santiago del Estero, Argentina.**
(DOCX)

## Acknowledgments

We would like to thank all the members of the local FMS office in Añatuya for their continuous compromise with the program, the Municipality of Añatuya, the Dirección Provincial de Enfermedades Transmitidas por Vectores (Santiago del Estero), the Instituto Nacional de Tecnología Agropecuaria (INTA) offices of Añatuya, Los Juríes and Santiago del Estero. We would also like to acknowledge support from the IT department of InSuD in the development of the study online platform, especially Mariano Caorsi and Javier Ornella. Fig 1 was elaborated with the help of Leonardo Sandon and Karina Cardone helped with the statistical analysis, both from Mundo Sano.

## Author Contributions

**Conceptualization:** Elsa Segura, Marcelo Claudio Abril.

**Data curation:** Diego Weinberg, Maria Florencia Casale, María Victoria Periago.

**Formal analysis:** Diego Weinberg, Maria Florencia Casale, María Victoria Periago.

**Funding acquisition:** Elsa Segura, Marcelo Claudio Abril.

**Investigation:** Diego Weinberg, Maria Florencia Casale, Rosa Graciela Cejas, María Victoria Periago, Elsa Segura, Marcelo Claudio Abril.

**Methodology:** Diego Weinberg, Maria Florencia Casale, Rafael Hoyos, María Victoria Periago, Marcelo Claudio Abril.

**Project administration:** Diego Weinberg, Maria Florencia Casale, Marcelo Claudio Abril.

**Resources:** Marcelo Claudio Abril.

**Supervision:** Diego Weinberg, Rosa Graciela Cejas, Rafael Hoyos, María Victoria Periago, Elsa Segura.

**Validation:** María Victoria Periago, Marcelo Claudio Abril.

**Visualization:** María Victoria Periago, Marcelo Claudio Abril.

**Writing – original draft:** Diego Weinberg, Maria Florencia Casale, María Victoria Periago.

**Writing – review & editing:** María Victoria Periago, Marcelo Claudio Abril.

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
