## [Decision Letter · Decision Letter 0]

20 Jan 2023

Dear Dr. Periago,

Thank you very much for submitting your manuscript "Chagas prevention and control in an endemic area from the Argentinian Gran Chaco Region: data from 14 years of uninterrupted intervention" for consideration at PLOS Neglected Tropical Diseases. As with all papers reviewed by the journal, your manuscript was reviewed by members of the editorial board and by several independent reviewers. In light of the reviews (below this email), we would like to invite the resubmission of a significantly-revised version that takes into account the reviewers' comments. 

This paper describes the effect of a long-term evolving intervention in the Argentine Chaco that decreased triatomine infestation over time. In addition to the comments mentioned by the reviewers I have a few suggestions:

- It would be helpful to define "ecohealth" in the introduction.

- Table 2 needs some explanation about why there are more houses improved than currently exist in some sites.

- If available, it would be interesting to report the prevalence of Chagas disease in specifically in children (esp. <5 years old) in these areas over the course of the intervention, as this more accurately reflects ongoing transmission.

- The discussion could benefit from addressing sustainability of the intervention and its applicability in other endemic places (Bolivia, etc).

We cannot make any decision about publication until we have seen the revised manuscript and your response to the reviewers' comments. Your revised manuscript is also likely to be sent to reviewers for further evaluation.

Sincerely,

Natalie Bowman, MD

Academic Editor

Esther Schnettler

Section Editor

This paper describes the effect of a long-term evolving intervention in the Argentine Chaco that decreased triatomine infestation over time. In addition to the comments mentioned by the reviewers I have a few suggestions:

- It would be helpful to define "ecohealth" in the introduction.

- Table 2 needs some explanation about why there are more houses improved than currently exist in some sites.

- If available, it would be interesting to report the prevalence of Chagas disease in specifically in children (esp. <5 years old) in these areas over the course of the intervention, as this more accurately reflects ongoing transmission.

- The discussion could benefit from addressing sustainability of the intervention and its applicability in other endemic places (Bolivia, etc).

Reviewer's Responses to Questions

**Key Review Criteria Required for Acceptance?**

**Methods**

-Are the objectives of the study clearly articulated with a clear testable hypothesis stated?

-Is the study design appropriate to address the stated objectives?

-Is the population clearly described and appropriate for the hypothesis being tested?

-Is the sample size sufficient to ensure adequate power to address the hypothesis being tested?

-Were correct statistical analysis used to support conclusions?

-Are there concerns about ethical or regulatory requirements being met?

Reviewer #1: (No Response)

Reviewer #2: Please refer to my comments below

Reviewer #3: Some lacking statistical analysis with respect to intra and peridomicile infestation rates should be conducted. See comments below

**Results**

-Does the analysis presented match the analysis plan?

-Are the results clearly and completely presented?

-Are the figures (Tables, Images) of sufficient quality for clarity?

Reviewer #1: (No Response)

Reviewer #2: Please refer to my comments below

Reviewer #3: No issues

**Conclusions**

-Are the conclusions supported by the data presented?

-Are the limitations of analysis clearly described?

-Do the authors discuss how these data can be helpful to advance our understanding of the topic under study?

-Is public health relevance addressed?

Reviewer #1: (No Response)

Reviewer #2: Please refer to my comments below

Reviewer #3: Discussion and conclusions should be worked on for clarity and detail addition. The study is relevant for public health and this is discussed in the manuscript.

**Editorial and Data Presentation Modifications?**

Reviewer #1: (No Response)

Reviewer #2: A thorough revision of the English is strongly suggested. Spanglish seems to occur (for ex. Line 214 : deposit instead of storerooms).

Minor comments

Please give the url to access reference 19. This reader could not find it on the web 

Ref 20, 45 and 43: I suggest including the corresponding urls to help accessing these references. 

L227: There is a typo: “but if not adult or nymph bugs are observed,”

Suppl File 3: Please explain the abbreviations included in this record sheet.

Reviewer #3: The manuscript can be enhanced by an English style revision.

**Summary and General Comments**

Reviewer #1: Dear authors,

Thank you for providing me with the opportunity to review this manuscript. This study presents data from the 14-year field engagement, which are highly worth being published. However, the current version of the manuscript reads more like a project report, rather than a scientific research paper. I encourage the authors to consider the following points:

A) What is your research question(s)?

B) Distinguish clearly between the observers (“the study”) and the observed (“the program”). It may be hard to separate them in this kind of work because the authors have been implementing and studying the program at the same time. However, at least conceptually, these two endeavors need to be distinguished.

C) Try to make some inferences, whether descriptive or causal. By inferences, I mean that we say something about unobserved facts using the facts we have observed.

Among many possible ways to improve this manuscript, the authors may choose:

1) Reframe the study as an assessment of the echohealth intervention. The study will assess the impact of the 14-year intervention on entomological indicators.

2) First, describe the contents and processes of interventions.

3) Second, describes the longitudinal trends in entomological indicators.

4) Then, discuss whether the observed changes (or sustained reductions) in entomological indicators can be attributable to the intervention. To what extent is the echohealth intervention effective in reducing vector populations?

5) If you have data, it may be helpful to compare the entomological index between the settlements with and without the echohealth intervention.

Please note that the authors do not have to exactly follow these guidelines from 1) to 5), but I hope they can give the authors some ideas to improve the manuscript.

Thank you.

Reviewer #2: The title of this article focus on data on the ongoing project but the data is not available. I suggest changing the title

and also including the relevant data as a supplementary material.

The authors state “Therefore, the objective of the current study is to show the evolution of the program” as this is a research article, the authors need to clarify which is the research question. For this reader describing the evolution of the program is not worth publishing in Plos NTD, especially if no figures on the costs invested in the program are included or no relation between the program described and the expected outcomes are inquired. No figures are included on the amount of effort in control actions, for instance insecticide spraying (number of houses sprayed, number of houses re-sprayed per community, etc). 

Abstract

Lines 26-28: The authors state “Argentina´s ChD program was originally vertical and 27 centralized; later, it was partially and unsuccessfully transferred to the provinces” Is this a generalized fact? How can the authors explain the fact that there are endemic provinces were success has been achieved (i. e.: measured as certification of vector-borne domestic T. cruzi interruption?

Lines 40-41: “Intra and peri-infestations” refer to intra-domiciliary and peridomestic infestation? I suggest the authors spell out these words for sake of clarity.

Author summary

L49-52: Please correct Triatoma is only the main vector in The Gran Chaco region and Arequipa. The authors need to narrow down their affirmations.

L56-58: Please revise the writing of this sentence “the most cases of ChD” sounds incorrect.

Introduction

L 99-101. “Despite these public health measures, the national control of ChD still eludes us due to inconsistency in the application of these measures, a problem that 101 transcends our borders “

It is not clear to this reader what is eluded. (“us” seems to be a mistake in this sentence).

L115-116 “Santiago del Estero is located within the Gran Chaco Region, which is historically one of the regions with the most cases of vector transmitted ChD [22, 23].” Please revise this expression.

L122 The term “ecotope” to refer to the monte forest seems inaccurate. 

L133 and 134: Ref 33 for these lines is incorrect. I suggest citing Gürtler et al., 2009. PNAS 104 (41) 16194-16199. https://doi.org/10.1073/pnas.0700863104

M&M 

I suggest using Past Tense

A timeline indicating when each of the activities took place and when each community entered the entomological program and the housing improvement program would be very helpful.

L 342-343. “The interior of the houses, roof improvement and waterproofing have remained the same to this day” What is the meaning of this phrase? These improvements were performed in 2005 and remained unchanged from that onwards or the type of improvement has not changed along the program? Please clarify.

Results

L366-368. Indeed in the Supl material Fig 4 included as an example, the community is asking for insecticide spraying (not entering to the program as stated in L366-368). Please rewrite these sentences for sake of accuracy.

Table 2 should be unified with Table 1

L 389-392. My main suggestion regarding results is to re-analyze and rewrite this paragraph and the following ones that refer to the entomological results. As they are right now, results from baseline infestations are joined with results from communities that may be in the surveillance phase just because field work took place at the same year. Thus, I suggest expressing results in reference to the number of cycles and not to the year when they took place as it is in Figure 3 but for all the communities involved. 

Is there an explanation for the abrupt increase in infestation levels observed in La Salamanca (Figure 3C)? It seems to occur in january 2010, i.e.: after massive insecticide spraying of all the community and after the housing improvement was launched? Who looked for triatomines in these houses? Has the same team performed triatomines searches in 2007 -2008 and 2010 in La Salamanca? 

Please include the number of houses sprayed with insecticides per year per community 

Please include de number of houses sprayed in figures 2 and 3 

L417-420 repeat information from M&M. I suggest deleting them.

Discussion

The impact of the program on T. cruzi transmission is not addressed in this manuscript. 

Conclusions:

L577-8:“The program described within has shown that the ecohealth approach is effective for long term surveillance and control of ChD.” At what cost? If this program is to replicated elsewhere, which is the input needed?

Reviewer #3: Comments to the authors

The work presented is very interesting and deals with a long term problem of the region. This integrated and community based management program is promising and already showing good results. With respect to the manuscript I encourage authors to look for some shortening of the text, which can be achieved by reorganizing the information and avoiding redundant phrases. It would be interesting to show statistical information regarding the comparison of peri and intra domestic infestation. The discussion section dedicates some lines to covering results. This is unnecessary and the discussion would be enriched by comparing the results of this program with other programs that included, or not, house improvements. This could be done by including more details on cited work.

Specific comments

Line 77-80 This seems unclear, please rephrase. I suggest in line 79 replacing “and currently….into” for “corresponding to”

Line 93 Mentioning “Almost 20 years later“ sounds awkward in this context, I suggest starting the line directly with “In 1980”

Lines 112-114. This is mentioned later in lines 147-150. Please organize the text for one mention only.

Lines 119-123 “More specifically…..Lived”. This looks more appropriate for the materials and methods section.

Lines 144-146 “With…..evolved” looks redundant. I suggest erasing this sentence and modifying the following, eg “By 2005 our program, that had started in urban areas, was expanded to….·

Line 148 Replace “the component…. was added” by “This was done” or similar. In general there are several mentions to “the improvement of households” so I encourage authors to look for ways for rephrasing the term. 

Line 150 Remove “individuals”

Line 173 -174 Is this total number of households? If so, please add “total”. Remove “in the settlement”. Please provide some details on the meaning of “previous relation with FMS” and “organization of the community”. The rest of the paragraph corresponds to Type of experimental design so the authors may rephrase the title of the section.

Line 206-207 The request for consent is mentioned in line 191. I suggest including that consent was requested in each visit in line 191, and erasing that in lines 206-207.

Line 219 Rephrase “If at least one T. infestans individual is found the house is considered positive, and details on the specimen /s found, such as place of discovery (…), life stage (…) and …. is recorded”. Then the line 223-224 “A house…. both” can be removed. If presence of other triatomine species determined positivity please mention. 

Lines 236-238 Because the collected information was mentioned earlier, this sentence could be shortened to saying that all the information gathered was included in the platform, and moved to Statistical analysis section.

Line 238-240 This sentence may fit better at the beginning of the section, when the overall process of the program is presented

Line 253 In a) you could say “positive houses” 

Lines 263-265 Is this the same information as in lines 250-251? If so, please remove one mention, otherwise please rephrase for clarity

Lines 265-271 Looks better suited for Study Area section.

Line 291 I suggest replacing “In order….. household” by “Secondly”

Line 295 I suggest replacing “ according….. defining” by “as”

Line 303 While some habits and building structures can facilitate the settlement of triatomine populations I would be cautious of saying they “lead” to infestation. Possibly a different word would better represent reality. 

Statistical analysis: In this section you could include a line mentioning which information was recorded and saved. Lines 360-361 Can you please be more specific on what was compared?

Lines 424-429 This description could be improved by making statistical comparisons. Are the insects statistically more frequent in peridomestic areas? Is some peridomestic area statistically more infested? Are the insects statistically more frequent in bedrooms than is other areas of the intra domicile?

Lines 451-459 Most of this information was mentioned earlier. Please revise and delete redundant information.

Line 468 Delete “these were taken…… herein”

Line 471-474 Starting at “different…” should be moved to the next paragraph. For example at line 483 delete “unlike this type of programs” and insert the previous sentence eg as this “Because different studies highlight …households, “and” tackling…..insufficient”, the experience described herein……

Line 502 Authors may be able to find details on infestation rates of other areas that were not part of this study, either at governmental Chagas disease programs or per request to other authors working in the topic (eg the group of Ricardo Gürtler). Showing some type of comparison with areas that only received insecticide treatment (and where no resistance to pyrethroids has been described) would be very interesting.

Lines 515-524 This part of the paragraphs is rather redundant with previous information appearing in the text. This could be reduced to one sentence that opens the following discussion on the ecohealth approach.

Lines 531-533 This is important, as sometimes the effectivity of insecticide spraying is decreased by inconsistent periods of application, insecticide resistance and the impossibility to access every house in a given area, among other issues. There is evidence that the residual effect of insecticides is different depending on the construction material (eg Germano et al 2014 “Fenitrothion….”, Gürtler et al 2004 “Effectiveness…”, Rojas de Arias et al 2004 “Pyrethroid…” and others) so house improvements may enhance the result of spraying by providing a different wall plastering. 

Lines 549-554 The failure of a program for eradicating ranchos was discussed earlier by lines 475 and on. Please reorganize the text for only one mention of this.

Lines 564-574 This paragraph is rather redundant with previous ones. Possibly it could be resumed in one or two sentences mentioning that further studies are required to determine the role of the settlement characteristics and of the natural areas in the differences of infestation between areas.

Supplementary material. Letter from Lote 27. The letter requests insecticide spraying. Did the community request other involvement in the program? Did any other community request participation?

PLOS authors have the option to publish the peer review history of their article (what does this mean?). If published, this will include your full peer review and any attached files.

Reviewer #1: No

Reviewer #2: No

Reviewer #3: No
---

## [Decision Letter · Decision Letter 1]

5 Apr 2023

Dear Dr. Periago,

Thank you very much for submitting your manuscript "Chagas prevention and control in an endemic area from the Argentinian Gran Chaco Region: data from 14 years of uninterrupted intervention" for consideration at PLOS Neglected Tropical Diseases. As with all papers reviewed by the journal, your manuscript was reviewed by members of the editorial board and by several independent reviewers. In light of the reviews (below this email), we would like to invite the resubmission of a significantly-revised version that takes into account the reviewers' comments. 

In addition to the edits requested by the reviewers, please submit both a tracked changes and clean version of the revised manuscript.

We cannot make any decision about publication until we have seen the revised manuscript and your response to the reviewers' comments. Your revised manuscript is also likely to be sent to reviewers for further evaluation.

Sincerely,

Natalie Bowman, MD

Academic Editor

Esther Schnettler

Section Editor

In addition to the edits requested by the reviewers, please submit both a tracked changes and clean version of the revised manuscript.

Reviewer's Responses to Questions

**Key Review Criteria Required for Acceptance?**

**Methods**

-Are the objectives of the study clearly articulated with a clear testable hypothesis stated?

-Is the study design appropriate to address the stated objectives?

-Is the population clearly described and appropriate for the hypothesis being tested?

-Is the sample size sufficient to ensure adequate power to address the hypothesis being tested?

-Were correct statistical analysis used to support conclusions?

-Are there concerns about ethical or regulatory requirements being met?

Reviewer #1: (No Response)

Reviewer #2: All my previous comments have been addressed

**Results**

-Does the analysis presented match the analysis plan?

-Are the results clearly and completely presented?

-Are the figures (Tables, Images) of sufficient quality for clarity?

Reviewer #1: (No Response)

Reviewer #2: (No Response)

**Conclusions**

-Are the conclusions supported by the data presented?

-Are the limitations of analysis clearly described?

-Do the authors discuss how these data can be helpful to advance our understanding of the topic under study?

-Is public health relevance addressed?

Reviewer #1: (No Response)

Reviewer #2: (No Response)

**Editorial and Data Presentation Modifications?**

Reviewer #1: (No Response)

Reviewer #2: (No Response)

**Summary and General Comments**

Reviewer #1: Dear authors,

Thank you for your efforts in revising the manuscript. As a general comment, I strongly recommend the authors review the manuscript carefully and thoroughly to avoid rejection in the next review round. I provide the list of specific comments below, with some points that must have been corrected by the authors before the authors asked reviewers to work. More importantly, please do not consider this list exhaustive: responding to these comments only may not be sufficient to improve the quality of writing to the acceptable level.

Line 99: What does “national control of ChD” mean? When can it be “achieved”?

Line 100: Who are “our”?

Line 104-5: It seems that this paper describes more than the S&C program, according to the Method section.

Line 129: “households” should be “houses,” if you are talking about physical structures.

Line 131: There is a logical leap. What connects settlements and serological studies?

Line 132-3: two? Two departments in the Province of Santiago del Estero?

Line 145: Check whether you talk about the S&C program only here.

Line 163-4: I don’t see any data or relevant arguments about access to diagnosis and treatment in this paper. This mention of diagnosis and treatment should probably be removed.

Table 1: “No. of settlements included” should be “Cumulative number of settlements included”

Table 1: “Current No. of completed S&C cycles” --- when is “current”? As of when?

Table 1: “Current No. of completed S&C cycles” --- it’s easier to read the table if “cycle” is explained in the table’s footnote (its definition appears far in Line 240).

Line 370-79: These sentences seem to merely describe Table 1, which is not necessary.

Line 391: Please consider the numbering of supplementary files and tables. I thought you were referring here to Table 2 in the main text.

Line 420: What are “findings”?

Line 449: The paper describes more than the S&C program.

Line 461-71: What is the point of this paragraph?

Line 472-89: What is the point of this paragraph?

Line 490: What is “this virtuous association”?

Line 519: What does “This” refer to?

Line 521: “were”?

Line 521-22: Which “individual and social determinants” were considered indeed?

Line 529: It’s not clear why the authors consider insecticide as a probable key driver to lowering vector infestation. Please make your arguments clearer.

Line 529-32: So, are you arguing here that insecticide alone is OK to control vector populations or are you insisting that insecticide should be combined with other measures? Make your claim clearer.

Line 533-549: These two paragraphs contain lots of serological data. Where do these data come from? If they come from the already published materials, please put citations. If they are original data of this paper, the authors should describe how these data were collected and analyzed in the Method section and report findings in the Result section.

Line 550: “knowledge” about what? What are “technological tools”?

Line 554: “access to technology” --- which technology are you talking about?

Line 561-9: What is the point of this paragraph? The first sentence is too long to figure out the authors’ main claim.

Line 570-81: What is the point of this paragraph? Are you trying to explain differences observed among settlements?

Discussion Section: As the authors pointed out in Introduction, many Latin American countries have experienced decentralization of vector control programs and moved towards PHC-integrated or community-based vector surveillance systems (e.g., see Hashimoto et al. BMC Health Serv Res. 2015; Yoshioka et al. Infect Dis Pov. 2017). After 14 years of implementing the S&C program in Argentina, do the authors insist that the Government of Argentina should hire field agents to maintain this kind of S&C program? Does it economically feasible and sustainable? If not, what could be an alternative?

Line 583-4: What do you mean by “the ecohealth approach is effective for long term [sic] S&C of ChD”? This sentence is confusing because S&C seems to be a component of the “ecohealth approach,” according to Lines 180-4. If so, S&C is not an outcome of the “ecohealth approach.”

Line 586: Where is the evidence for “< than [sic] 1% seroprevalence”? Data are not properly presented for this claim.

Line 584-587: How do all of the evidence guide control programs? What are the authors’ specific recommendations?

Line 593-4: “has enabled the population to get access to diagnosis and etiological treatment of ChD” --- this claim is not supported by data or relevant arguments in the paper.

Reviewer #2: All my previous comments have been addressed

Minor comment: 

The url of reference N 64 seems not to work. Please verify

PLOS authors have the option to publish the peer review history of their article (what does this mean?). If published, this will include your full peer review and any attached files.

Reviewer #1: No

Reviewer #2: No
---

## [Editor Report · Decision Letter 2]

23 May 2023

Dear Dr. Periago,

We are pleased to inform you that your manuscript 'Chagas prevention and control in an endemic area from the Argentinian Gran Chaco Region: data from 14 years of uninterrupted intervention' has been provisionally accepted for publication in PLOS Neglected Tropical Diseases.

Best regards,

Natalie Bowman, MD

Academic Editor

Esther Schnettler

Section Editor

Please proofread the manuscript carefully, as there are several punctuation and minor spelling mistakes. A few I noticed were (by line, not a complete list):

- 53: delete comma after "Even though"

- 127: change XIX to nineteenth

- 129: not clear what "lifted" means. Completed? Extended?

- 134: con should be on

- 293: change are to were

- 357: year should be years

- 358: remove comma after time

- 402: change to decreasing to

- 416: change were to where

-480: remove comma after herein

---

## [Editor Report · Acceptance letter]

1 Jun 2023

Dear Dr. Periago,

We are delighted to inform you that your manuscript, "Chagas prevention and control in an endemic area from the Argentinian Gran Chaco Region: data from 14 years of uninterrupted intervention," has been formally accepted for publication in PLOS Neglected Tropical Diseases.

Best regards,

Shaden Kamhawi

co-Editor-in-Chief

Paul Brindley

co-Editor-in-Chief
